# Opposing functions of the plant *TOPLESS* gene family during SNC1-mediated autoimmunity

**Christopher M. Garner**[1,2,3¤a], **Benjamin J. Spears**[1,3¤b], **Jianbin Su**[1,3], **Leland J. Cseke**[1,3], **Samantha N. Smith**[1,3], **Conner J. Rogan**[2,3¤c], **Walter Gassmann**[1,3]*

**1** Division of Plant Sciences, University of Missouri, Columbia, Missouri, United States of America, **2** Division of Biological Sciences, University of Missouri, Columbia, Missouri, United States of America, **3** Christopher S. Bond Life Sciences Center and Interdisciplinary Plant Group, University of Missouri, Columbia, Missouri, United States of America

¤a Current address: Thermo Fisher Scientific, Carlsbad, California, United States of America
¤b Current address: Butler University, Indianapolis, Indiana, United States of America
¤c Current address: Oregon State University, Corvallis, Oregon, United States of America
* gassmannw@missouri.edu

## Abstract

Regulation of the plant immune system is important for controlling the specificity and amplitude of responses to pathogens and in preventing growth-inhibiting autoimmunity that leads to reductions in plant fitness. In previous work, we reported that SRFR1, a negative regulator of effector-triggered immunity, interacts with SNC1 and EDS1. When *SRFR1* is nonfunctional in the Arabidopsis accession Col-0, *SNC1* levels increase, causing a cascade of events that lead to autoimmunity phenotypes. Previous work showed that some members of the transcriptional co-repressor family TOPLESS interact with SNC1 to repress negative regulators of immunity. Therefore, to explore potential connections between *SRFR1* and *TOPLESS* family members, we took a genetic approach that examined the effect of each *TOPLESS* member in the *srfr1* mutant background. The data indicated that an additive genetic interaction exists between *SRFR1* and two members of the *TOPLESS* family, *TPR2* and *TPR3*, as demonstrated by increased stunting and elevated *PR2* expression in *srfr1 tpr2* and *srfr1 tpr2 tpr3* mutants. Furthermore, the *tpr2* mutation intensifies autoimmunity in the auto-active *snc1-1* mutant, indicating a novel role of these *TOPLESS* family members in negatively regulating *SNC1*-dependent phenotypes. This negative regulation can also be reversed by overexpressing *TPR2* in the *srfr1 tpr2* background. Similar to TPR1 that positively regulates *snc1-1* phenotypes by interacting with SNC1, we show here that TPR2 directly binds the N-terminal domain of SNC1. In addition, TPR2 interacts with TPR1 *in vivo*, suggesting that the opposite functions of TPR2 and TPR1 are based on titration of SNC1-TPR1 complexes by TPR2 or altered functions of a SNC1-TPR1-TPR2 complex. Thus, this work uncovers diverse functions of individual members of the TOPLESS family in Arabidopsis and provides evidence for the additive effect of transcriptional and post-transcriptional regulation of *SNC1*.

**Data Availability Statement:** All relevant data are within the manuscript and its Supporting Information files.

**Funding:** This research was funded by a University of Missouri Life Sciences Graduate Fellowship (CMG), a Division of Plant Sciences Daniel F. Millikan Fellowship (BJS), the Life Sciences Undergraduate Research Opportunity Fellows Program (CJR), and National Science Foundation (nsf.gov) grant IOS-1456181 (WG). The funders had no role in study design, data collection and analysis, decision to publish, or preparation of the manuscript.

**Competing interests:** The authors have declared that no competing interests exist.

## Author summary

The immune system is a double-edged sword that affords organisms with protection against infectious diseases but can also lead to negative effects if not properly controlled. Plants only possess an innate antimicrobial immune system that relies on rapid upregulation of defenses once immune receptors detect the presence of microbes. Plant immune receptors known as resistance proteins play a key role in rapidly triggering defenses if pathogens breach other defenses. A common model of unregulated immunity in the reference Arabidopsis variety Columbia-0 involves a resistance gene called *SNC1*. When the SNC1 protein accumulates to unnaturally high levels or possesses auto-activating mutations, the visible manifestations of immune overactivity include stunted growth and low biomass and seedset. Consequently, expression of this gene and accumulation of the encoded protein are tightly regulated on multiple levels. Despite careful study the mechanisms of *SNC1* gene regulation are not fully understood. Here we present data on members of the well-known TOPLESS family of transcriptional repressors. While previously characterized members were shown to function in indirect activation of defenses, TPR2 and TPR3 are shown here to function in preventing high defense activity. This study therefore contributes to the understanding of complex regulatory processes in plant immunity.

## Introduction

Plants defend against infection by having a multilayered immune system, one branch of which recognizes molecular signatures of microbes through pattern recognition receptors at the cell surface. At the same time, plants monitor potential intracellular targets of pathogen attack [1,2]. At the heart of this intracellular plant surveillance system are the resistance genes of the nucleotide binding site–leucine-rich repeat (NLR) class [3]. Resistance proteins recognize, directly or indirectly, the actions of pathogen-secreted effector proteins which seek to interfere with plant immune responses or normal plant physiology. Upon sensing the activity of effectors, resistance proteins elicit a rapid and robust defense response, called effector-triggered immunity (ETI). In the case of the biotrophic defense response, this includes accelerated production of high levels of the plant hormone salicylic acid (SA) and the induction of *PATHO-GENESIS RELATED* (*PR*) genes [1].

Because of cross-talk between plant hormone pathways, activation of the defense response is accompanied by repression of pathways that promote growth [4–7]. Therefore, induction of the plant immune system must be kept under tight control to avoid fitness penalties incurred during the absence of pathogen infection [8], as illustrated by autoimmune mutants of Arabidopsis that display the negative effects of an unregulated immune response. More than thirty different mutants have been identified that cause an autoimmune response exhibited by dwarfism, high levels of salicylic acid, constitutive defense gene expression, and subsequent increased resistance to pathogens [9]. Genetic analysis of these mutants has provided a wealth of information regarding the identity of positive and negative regulators of the immune response, and they illustrate the many levels of regulation that take place within the plant immune system.

SUPRESSOR of rps4-RLD1 (SRFR1) is a negative regulator of ETI mediated by several NLR proteins with a Toll/interleukin-1 receptor domain at their N-termini (TNLs), including RPS4/RRS1 and SNC1 [10–12]. It was discovered in a genetic screen for mutants that were resistant to *Pseudomonas syringae pv. tomato* strain DC3000 expressing the bacterial effector

AvrRps4 in the Arabidopsis accession RLD, which is normally susceptible because of natural inactivating polymorphisms in the *RPS4* resistance gene [10]. Mutants of *srfr1* in the Col-0 background constitutively activate SNC1 expression, causing an autoimmune phenotype characterized by high levels of salicylic acid, constitutive expression of *PR* genes, and severe stunting [12,13]. This autoimmune phenotype is absent in the RLD background due to an absence of a full-length *SNC1* allele [12]. SRFR1 interacts with the TNLs RPS4, RPS6, and SNC1 as well as the central ETI regulator EDS1 in a complex disrupted by AvrRps4 [2,14]. Furthermore, *srfr1 eds1* mutants lose increased resistance phenotypes [14]. These results place SRFR1 as a key regulator of effector-triggered immunity conferred by the TNL class of resistance genes.

In addition to interactions within an ETI protein complex, homology to transcriptional regulators and interaction with transcription factors suggest SRFR1 could also be part of a transcriptional repressor complex [11]. SRFR1 interacts with members of the TEOSINTE BRANCHED1-CYCLOIDEA-PROLIFERATING CELL FACTOR (TCP) transcription factor family in the nucleus. Specifically, SRFR1 interacts strongly with TCP8, TCP14, and TCP15, and a triple *tcp8 tcp14 tcp15* mutant is compromised in effector-triggered immunity [15]. This interaction between SRFR1 and positive ETI regulators suggests a model wherein SRFR1 is restricting TCP access to promoters of defense-related genes, or recruiting other proteins that function as repressors of transcription at these promoters.

The five member Arabidopsis *TOPLESS* gene family (*TPL*, *TOPLESS RELATED1*, *TPR2*, *TPR3*, and *TPR4*) encodes members of the larger GRO/TUP1 family of corepressors that are proposed to interact with DNA-binding proteins in the promoter regions of regulated genes to repress transcription [16]. Analysis of TPL/TPR family interactions with transcription factors indicates that they have been coopted multiple times to regulate gene expression in diverse processes, including control of flowering time, hormone signaling, and stress responses [17]. Structural studies also provide evidence that TPL tetramerizes as part of its interactions with protein partners, suggesting the possibility of heterotetramers within the TOPLESS family [18].

Furthermore, TPR1 was shown to interact with SNC1, and together the complex, with an as yet unknown DNA-binding transcription factor, represses transcription of genes that function as negative regulators of defense responses such as *DEFENSE NO DEATH 1* (*DND1*) and *DND2*, which encode cyclic nucleotide-gated ion channels [19,20]. Therefore, similar to the interactions of SRFR1 with the TNL-mediated ETI machinery and transcription factors, TOPLESS family members display multiple mechanisms in their functions as co-repressors.

Whether SRFR1 is acting as part of a complex with the ETI machinery or functions as a transcriptional co-repressor, which molecular pathways regulate the autoimmunity phenotype of *srfr1* mutants remains a pressing question. Both models presented us with the possibility that *SRFR1* may also be interacting, at least genetically, with members of the TOPLESS family. Thus, we hypothesized that loss-of-function mutations in the *TOPLESS* gene family in the *srfr1-4* background would display similar phenotypes to the *tpl/tpr1* mutants in the *snc1-1* auto-active mutant background, reducing the *SNC1*-mediated autoimmune response. Here, we report the unexpected result that mutations in *TPR2* and *TPR3* have the opposite effect from those in *TPR1*, increasing the *SNC1* autoimmune response in the *srfr1-4* mutant background. This presents a novel function for TPR2 and TPR3 in either repressing positive regulators of the immune response or interfering with the SNC1-TPR1-mediated repression of negative regulators.

## Results

### Mutations in *TPR2* exacerbate the *srfr1-4* autoimmune phenotype

To investigate possible genetic interactions between *SRFR1* and members of the *TOPLESS* family, *srfr1-4* was crossed with T-DNA mutants in *TPL*, *TPR1*, *TPR2*, *TPR3*, and *TPR4*.

Homozygous *srfr1-4 tpl/tpr* double mutants were compared to *srfr1-4* to determine if stunting, a measure of constitutively activated defenses, was affected. To quantify these differences in stunting we also measured shoot weights from each genotype after 4 weeks of growth. The results showed that *srfr1-4 tpl-8* and *srfr1-4 tpr2-2* were significantly different from *srfr1-4* in terms of size and overall shoot mass, in opposite directions ([Fig 1]). No difference in shoot

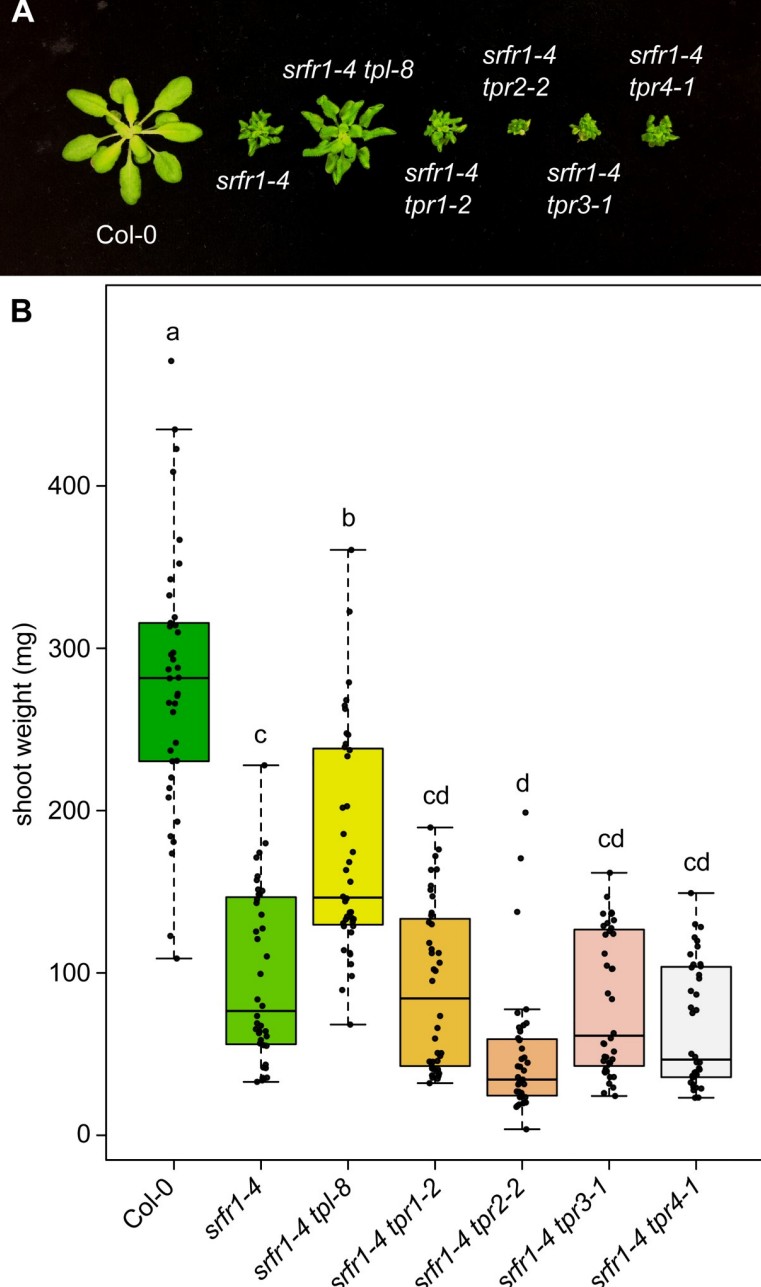

**Fig 1. Loss of function of *TPR2* increases stunting in *srfr1*.** (A) Morphological phenotype of *srfr1-4* and *srfr1-4 tpl/ tpr* double mutants at four weeks post sowing. (B) Shoot weight from plants grown under short day conditions at 21˚C for four weeks. Dots represent individual data points taken over two separate experiments. Whiskers on boxplots are drawn to the farthest data point within 1.5 * IQR of first and third quartiles. Letters denote significant differences as determined by Student's t-test ($P<0.01$) using the Bonferroni-Holm method to correct for multiple comparisons.

mass was observed in the single *tpl/tpr* mutants compared to Col-0 (S1 Fig). *PR2* is well established as an overall marker of immune system activation, and we found that the degree of stunting in this panel of auto-immune mutants correlated with their level of PR2 expression (S2 Fig).

Stunting in *srfr1-4* is due to the activation of the TNL gene *SNC1* [13,19]. Given that it was shown that mutation of *tpl* lessens the effect of stunting in autoactive *snc1-1* mutants [19], and the dependency of stunting in *srfr1-4* on activation of *SNC1*, we concluded that the effect we were seeing in *srfr1-4 tpl-8* mutants was a recapitulation of previous findings and chose not to investigate this mutant further. We did not see a similar phenotype in *srfr1-4 tpr1-2*. The T-DNA insertion in the *tpr1-2* allele occurred in an intron in the 5' untranslated region, and the absence of a phenotype indicates that this allele is not a complete knockout. In contrast, the increased stunting of *srfr1-4 tpr2-2* represents a novel genetic interaction, and as such we switched our focus to concentrate on the *SRFR1-TPR2* interaction. Stunting was alleviated in the *srfr1-4* single and the *srfr1-4 tpl/tpr* double mutants when plants were grown at higher temperature, consistent with growth phenotypes of other mutants with activated SNC1 [12]. This indicates that the increased stunting observed with *srfr1-4 tpr2-2* is also dependent on *SNC1* (S3 Fig).

To verify that the increased autoimmunity phenotype was indeed caused by the insertion at the *TPR2* locus and not some other tightly linked mutation, we obtained a second allele of *TPR2*, *tpr2-1*, and crossed this allele to *srfr1-4*. For both *tpr2* alleles we did not detect *TPR2* mRNA (S4 Fig). As with *srfr1-4 tpr2-2*, we saw increased stunting in the *srfr1-4 tpr2-1* double mutant relative to *srfr1-4* (Fig 2A). To quantify these differences in stunting we measured shoot weights from each genotype after 4 weeks of growth. The results showed that *srfr1-4 tpr2-1* and *srfr1-4 tpr2-2* were significantly different from *srfr1-4* in terms of overall shoot mass (Fig 2B), but that neither *TPR2* single mutant was significantly different from Col-0.

## *TPR2* and *TPR3* are partially redundant or function additively in repressing autoimmunity in *srfr1-4*

Previous research has demonstrated functional redundancy amongst TOPLESS family members, and that higher order *tpl/tpr* knockouts produce stronger phenotypes than single *tpl/tpr* mutants [21–23]. Based on the close evolutionary relatedness of *TPR2* and *TPR3* (S5 Fig) and previous reports that indicated *TPL*, *TPR1*, and *TPR4* are repressors of negative regulators of immunity [19], we chose to investigate if mutations in *TPR3* would impact the *srfr1-4 tpr2-2* phenotype. To obtain a *srfr1-4 tpr2-2 tpr3-1* triple mutant, *srfr1-4 tpr2-2* was crossed with *srfr1-4 tpr3-1*. Analysis of shoot mass showed that the *srfr1-4 tpr2-2 tpr3-1* triple mutant is significantly smaller than both *srf1-4* and *srfr1-4 tpr2-2* (Fig 3A and 3B).

As TOPLESS family members have been shown to be repressors of transcription we decided to examine the mRNA levels of *SNC1* in the *srfr1-4 tpr2-2* and *srfr1-4 tpr2-2 tpr3-1* mutants to see if they were affected relative to *srfr1-4*. We also examined *PR2* expression as a marker for overall immune activation and used qPCR rather than protein blotting to quantify subtle differences in mRNA levels for the remainder of this study. As illustrated in Fig 3C and 3D, *PR2* and *SNC1* mRNA levels were significantly increased in *srfr1-4 tpr2-2* and *srfr1-4 tpr2-2 tpr3-1* relative to *srfr1-4*; however, no significant change in *PR2* or *SNC1* expression was observed in the *tpr2-2* or *tpr3-1* single mutants.

Given the partial redundancy or additive function observed between *TPR2* and *TPR3* in the *srfr1-4* background and the lack of any observable phenotype in the single mutants, we crossed *tpr2-2* to *tpr3-1* to create a *tpr2-2 tpr3-1* double mutant. No stunting or other morphological phenotypes were observed in *tpr2-2 tpr3-1* (Fig 4A). We also found no significant difference

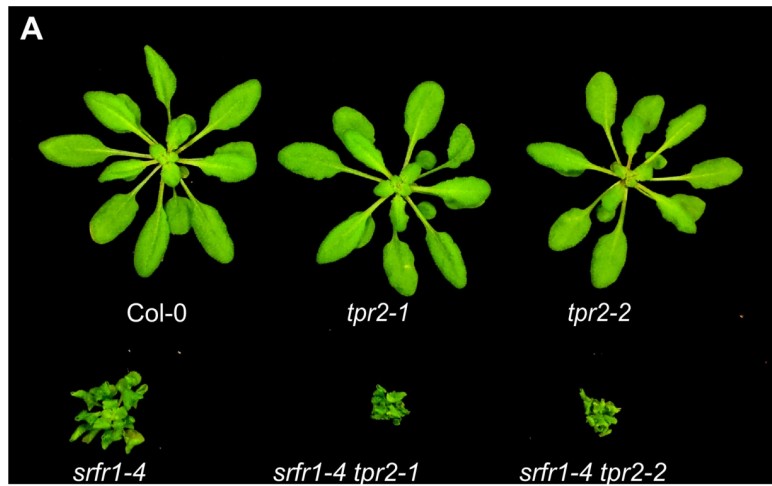

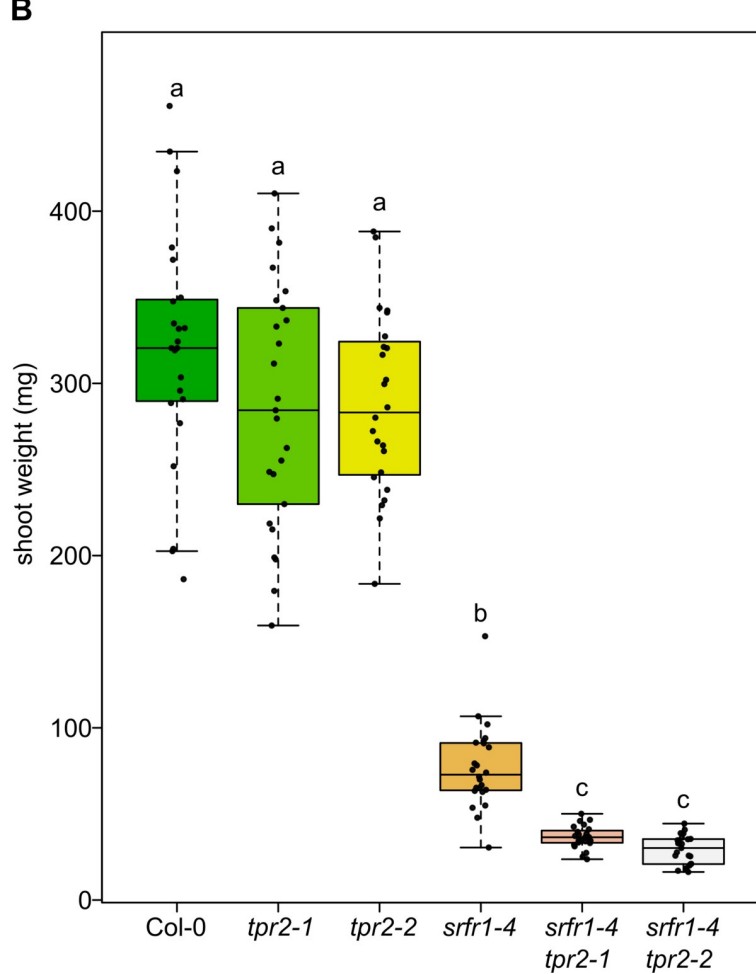

**Fig 2. Multiple alleles of *TPR2* increase stunting in *srfr1*.** (A) Morphological phenotypes of *tpr2-1*, *tpr2-2*, *srfr1-4*, *srfr1-4 tpr2-1*, and *srfr1-4 tpr2-2* at four weeks post sowing. (B) Shoot weight from plants grown under short day conditions at 21˚C for four weeks. Dots represent individual data points. Whiskers on boxplots are drawn to the farthest data point within 1.5 * IQR of first and third quartiles. Letters denote significant differences as determined by Student's t-test ($P<0.05$) using the Bonferroni-Holm method to correct for multiple comparisons.

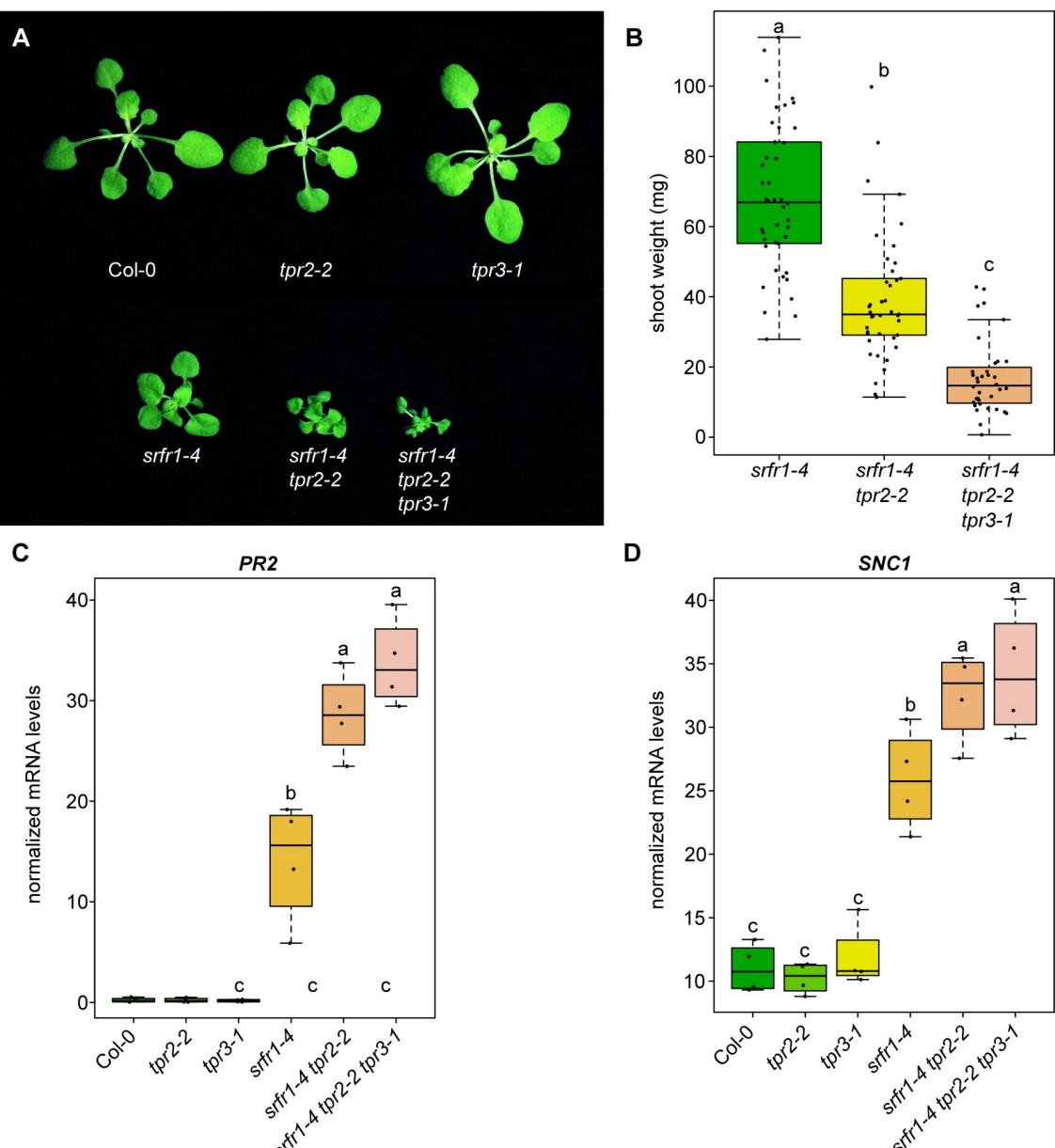

**Fig 3. Simultaneous loss of *TPR2* and *TPR3* increases stunting and expression of *PR2* and *SNC1* in *srfr1*.** (A) Morphological phenotype of *srfr1-4*, *srfr1-4 tpr2-2*, and *srfr1-4 tpr2-2 tpr3-1* at 20 days after sowing. Plants were grown under short day conditions at 21˚C. (B) Shoot weight from plants grown under short day conditions at 21˚C for four weeks. Dots represent individual data points taken over two separate experiments. Whiskers on boxplots are drawn to the farthest data point within 1.5 * IQR of first and third quartiles. Letters denote significant differences as determined by Student's t-test ($P<0.001$) using the Bonferroni-Holm method to correct for multiple comparisons. (C&D) Expression as measured by quantitative RT-PCR of *PR2* and *SNC1* in single, double, and triple mutants. Dots represent individual data points taken over two separate experiments. Genes of interest were normalized against *SAND* (At2g28390). Whiskers on boxplots are drawn to the farthest data point within 1.5 * IQR of first and third quartiles. Letters denote significant differences as determined by Student's t-test ($P<0.05$) using the Bonferroni-Holm method to correct for multiple comparisons.

between Col-0 and *tpr2-2 tpr3-1* with regards to *PR2* expression; however, we did see a small but significant increase in *SNC1* expression in *tpr2-2 tpr3-1* when compared to Col-0 (Fig 4B and 4C). Consistent with the molecular data we did not observe a difference in resistance to DC3000 in *tpr2-2*, *tpr3-1*, or *tpr2-2 tpr3-1* compared to Col-0 (S6 Fig).

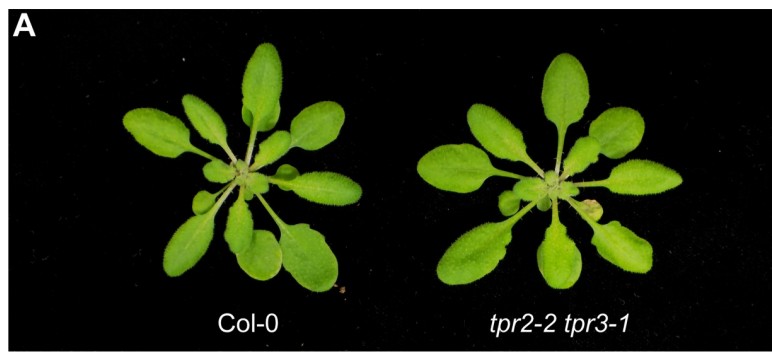

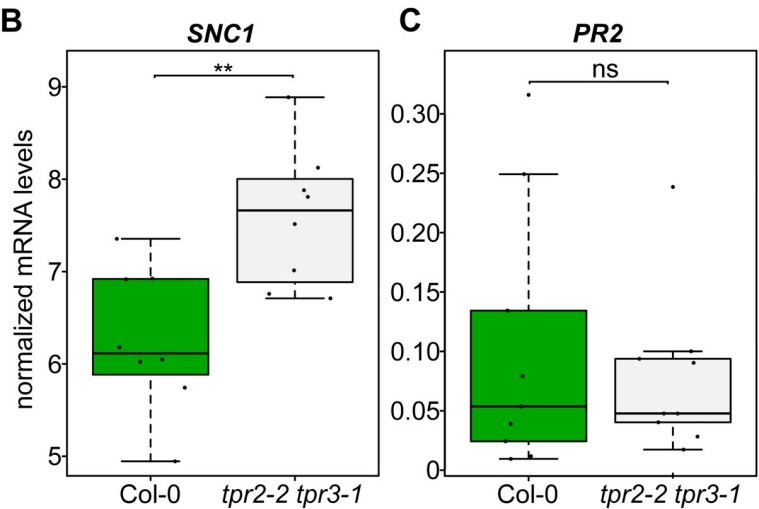

**Fig 4. *SNC1* expression is increased in *tpr2 tpr3*.** (A) Morphological phenotype of *tpr2-2 tpr3-1*. Plants were grown for four weeks under short day conditions at 21˚C. (B&C) Expression as measured by quantitative RT-PCR of *PR2* and *SNC1*. Dots represent individual data points taken over two separate experiments. Genes of interest were normalized against *SAND* (At2g28390). Whiskers on boxplots are drawn to the farthest data point within 1.5 * IQR of first and third quartiles. Asterisks denote significant differences as determined by Student's t-test (*P*<0.005) using the Bonferroni-Holm method to correct for multiple comparisons.

## Overexpression of *TPR2* in the *srfr1-4* background represses autoimmunity

We next asked if overexpressing *TPR2* would have the opposite effect and suppress autoimmunity in the *srfr1-4 tpr2-2* background. To test this hypothesis we cloned the *TPR2* coding sequence as a translational fusion with a C-terminal 10xMyc tag behind the constitutively active cauliflower mosaic virus 35S promoter. Using the 35S:*TPR2-myc* construct, several stable lines were created in the *srfr1-4 tpr2-2* genetic background. Two independent homozygous *TPR2-myc srfr1-4 tpr2-2* lines in the T3 generation were planted alongside Col-0, *srfr1-4*, and *srfr1-4 tpr2-2* to compare the degree of stunting. At four weeks after planting, the *TPR2-myc srfr1-4 tpr2-2* plants were less stunted than both *srfr1-4 tpr2-2* and *srfr1-4* (Fig 5A).

Quantification of *SNC1* showed that not only was transcript level reduced below *srfr1-4 tpr2-2* levels, but was also less than *SNC1* levels in *srfr1-4* (Fig 5B), correlating with plant size (Fig 5A). The TNL gene *RPP4* is located within the *SNC1* locus and has been shown to be co-regulated with *SNC1* both at the level of transcription and after transcription by RNA silencing [24]. We have also previously shown that *RPP4* is upregulated in *srfr1-4* [12]. To determine if *TPR2* also affects *RPP4* expression in the *srfr1-4* background, we quantified *RPP4* mRNA in *srfr1-4 tpr2-2* and in *TPR2-myc srfr1-4 tpr2-2*. We saw a slight non-significant increase in

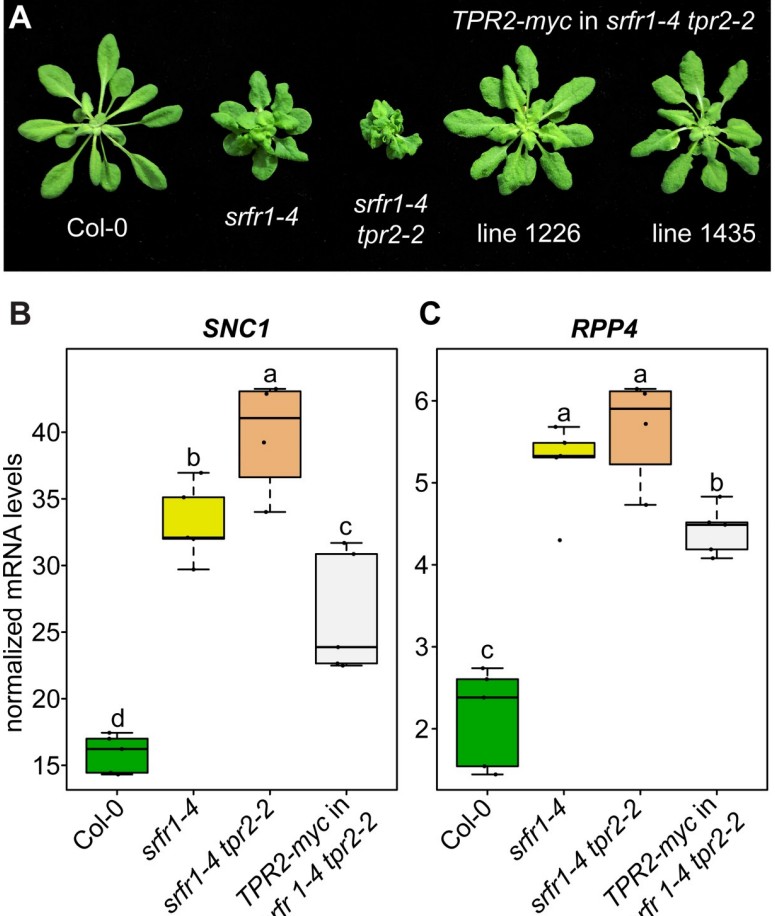

**Fig 5. Overexpression of TPR2 reduces stunting and *SNC1* expression in *srfr1 tpr2*.** (A) Morphological phenotype of *TPR2-myc srfr1-4 tpr2-2* compared to *srfr1-4* and *srfr1-4 tpr2-2*. Plants were grown under short day conditions at 21°C for four weeks. (B&C) Expression as measured by quantitative RT-PCR of *SNC1* and *RPP4*. Dots represent individual data points taken over two separate experiments. Genes of interest were normalized against *SAND* (At2g28390). Whiskers on boxplots are drawn to the farthest data point within 1.5 * IQR of first and third quartiles. Letters denote significant differences as determined by Student's t-test (*P*<0.05) using the Bonferroni-Holm method to correct for multiple comparisons.

*RPP4* expression in *srfr1-4 tpr2-2* relative to *srfr1-4*, while *RPP4* mRNA was reduced in *TPR2-myc srfr1-4 tpr2-2* below levels in *srfr1-4* (Fig 5C).

## Increased autoimmunity in *srfr1-4 tpr2-2* is partially dependent upon *SNC1*

Previous work has shown that stunting in *srfr1-4* is dependent on *SNC1*, and that a *srfr1-4 snc1-11* double mutant is morphologically normal but still expresses higher than normal levels of several defense-related genes [12]. To see if the enhanced autoimmunity that results from mutating *TPR2* in the *srfr1-4* background is dependent on *SNC1*, we created a quadruple mutant by crossing the *SNC1* knockout allele, *snc1-11*, to *srfr1-4 tpr2-2 tpr3-1*. As was previously observed for *srfr1-4 snc1-11*, we saw no stunting or morphological abnormalities in the *srfr1-4 snc1-11 tpr2-2 tpr3-1* quadruple mutant (Fig 6A). *SRFR1* regulation of *RPP4* is *SNC1* independent as *RPP4* is upregulated equally in both *srfr1-4* and *srfr1-4 snc1-11* relative to wild type levels in Col-0 [12]. Interestingly, *RPP4* expression was significantly decreased both in *srfr1-4 snc1-11 tpr2-2 tpr3-1* compared to *srfr1-4 snc1-11* and in *snc1-11 tpr2-2 tpr3-1*

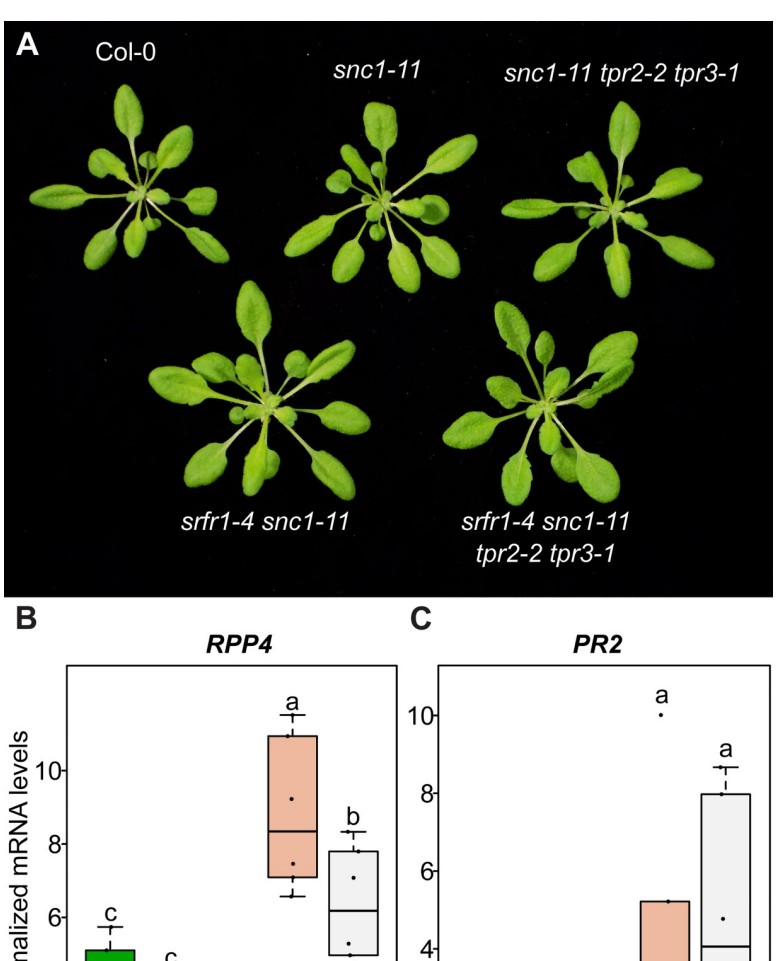

**Fig 6. *tpr2 tpr3* mutants have lower expression of *RPP4* in *snc1* knockouts.** (A) Morphological phenotype of plants harboring the *snc1-11* mutation crossed into *srfr1* and *tpr2 tpr3* mutants. Plants were grown under short day conditions at 21°C for four weeks. (B&C) Expression as measured by quantitative RT-PCR of *RPP4* and *PR2*. Dots represent individual data points taken over two separate experiments. Genes of interest were normalized against *SAND* (At2g28390). Whiskers on boxplots are drawn to the farthest data point within 1.5 * IQR of first and third quartiles. Letters denote significant differences as determined by Student's t-test ($P<0.05$) using the Bonferroni-Holm method to correct for multiple comparisons.

compared to *snc1-11* (Fig 6B), whereas *RPP4* mRNA levels in the *srfr1-4 tpr2-2* mutant were slightly higher than in *srfr1-4* (Fig 5C), indicating that these higher *RPP4* mRNA levels are at least partially dependent upon *SNC1*. Consistent with our previous study, we saw an increase in *PR2* levels in the *srfr1-4 snc1-11* double mutant compared to Col-0 and *snc1-11*. *PR2* levels

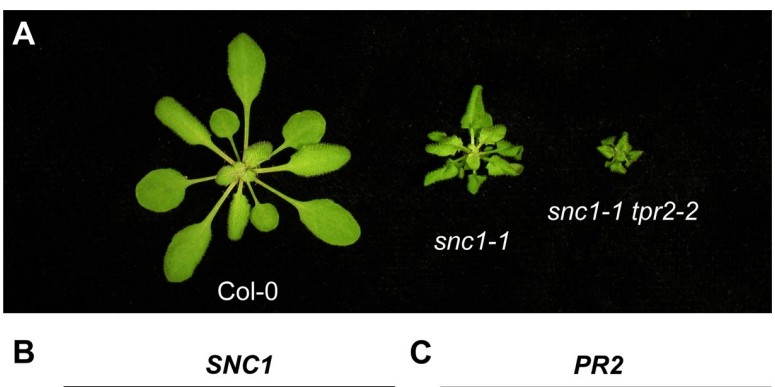

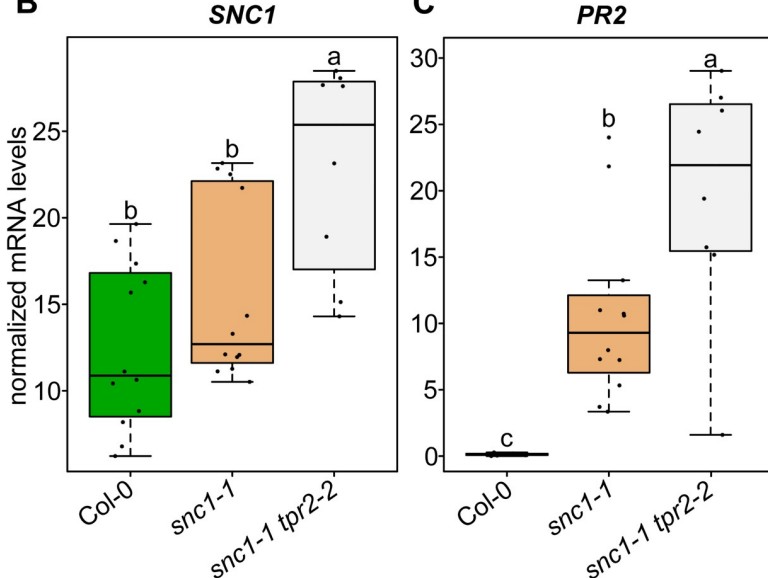

**Fig 7. Mutations in *TPR2* increase stunting and *SNC1* expression in *snc1-1* mutants.** (A) Morphological phenotypes of *snc1-1* and *snc1-1 tpr2-2*. Plants were grown under short day conditions at 21˚C for four weeks. (B&C) Expression as measured by quantitative RT-PCR of *SNC1* and *PR2*. Dots represent individual data points taken over two separate experiments. Genes of interest were normalized against *SAND* (At2g28390). Whiskers on boxplots are drawn to the farthest data point within 1.5 * IQR of first and third quartiles. Letters denote significant differences as determined by Student's t-test (*P*<0.01) using the Bonferroni-Holm method to correct for multiple comparisons.

in *srfr1-4 snc1-11 tpr2-2 tpr3-1* were comparable to those in *srfr1-4 snc1-11* (Fig 6C), and *in planta* bacterial growth assays showed comparable levels of increased resistance (S7 Fig).

To further investigate the relationship between *TPR2* and *SNC1* activity, we crossed *tpr2-2* to *snc1-1*, an auto-active allele of *SNC1* that induces a constitutive defense response and associated stunting [25]. The F2 from this cross produced approximately 1/16[th] plants which genotyped as homozygous *snc1-1 tpr2-2* that were extremely stunted and produced very little seed. When compared to *snc1-1*, *snc1-1 tpr2-2* was significantly more stunted, and had significantly higher levels of *SNC1* and *PR2* mRNA (Fig 7). These results are consistent with the conclusion that the autoimmune phenotypes modulated by mutations in *SRFR1* and *TPR2* are tightly associated with *SNC1*.

### *SRFR1* acts upstream of *SNC1* transcription

Transcription of *SNC1* is subject to feedback regulation through the production of salicylic acid. Upon activation of SNC1, SA accumulates in the plant and increased levels of SA cause even more transcription of *SNC1* [26]. Our data show that *tpr2-2* increases *SNC1* mRNA levels

in the *srfr1-4* and *snc1-1* backgrounds, but because of the complex feedback regulation of *SNC1* transcription it is unclear whether *SRFR1* and *TPR2* are directly affecting transcription at the *SNC1* locus, or if they are repressing some component downstream of SNC1 activation. Signaling for all Arabidopsis TNL class resistance proteins identified to date is dependent upon EDS1 [27], and mutating *EDS1* blocks the feedback regulation of SNC1, thereby making it possible to disambiguate events upstream of *SNC1* transcription from events downstream of SNC1 activation [28]. The *eds1-2* allele is a knockout for *EDS1* introgressed into Col-0 [29]. Previous work has shown that a *srfr1-4 eds1-2* double mutant shows no signs of enhanced basal resistance and is morphologically indistinguishable from Col-0 [14].

To determine if the *tpr2-2* mutation had any effect on transcription of *SNC1* in *srfr1-4 eds1-2*, we crossed *eds1-2* to *tpr2-2* and *srfr1-4 tpr2-2* to *srfr1-4 eds1-2* and obtained *eds1-2 tpr2-2* and *srfr1-4 eds1-2 tpr2-2* mutants. As seen previously with the *srfr1-4 eds1-2* double mutant, the *srfr1-4 eds1-2 tpr2-2* triple mutant was not morphologically different from Col-0 (Fig 8A). When we quantified the amount of *SNC1* transcript in these plants we found that *srfr1-4 eds1-2* produced significantly more *SNC1* than Col-0, *eds1-2*, and *eds1-2 tpr2-2* (Fig 8B). The *srfr1-4 eds1-2 tpr2-2* triple mutant had a repeatable but non-significant decrease in *SNC1* relative to *srfr1-4 eds1-2* (Fig 8B). These data suggest that *SRFR1* also acts upstream of *SNC1* transcription, while *TPR2* acts downstream of *SNC1* transcription.

TPR1 was previously shown to directly interact with the TIR domain of SNC1 [19]. To determine if TPR2 interacts with SNC1, we performed an *in vitro* pull down assay between GST-tagged TPR2 and T7-tagged SNC1-TIR domain. Pull down of GST-TPR2 with GST beads co-precipitated T7-SNC1-TIR, whereas pull down of GST alone failed to co-precipitate T7-SNC1-TIR (Fig 8C), indicative of a direct protein-protein interaction between TPR2 and SNC1. TOPLESS family members are not known to heteromerize except for TPR1 and TPR4 [17]. We therefore thought it likely that the post-transcriptional activity of TPR2 consists of competing with TPR1 for binding of SNC1. To test this *in vivo*, we transiently expressed GFP-SNC1, myc-TPR1 and HA-TPR2 in *Nicotiana benthamiana eds1-1* plants to minimize tissue disintegration by SNC1 activity [30]. Consistent with the *in vitro* data, both TPR1 and TPR2 were co-immunoprecipitated with SNC1 (Fig 9A). Interestingly, we discovered that TPR1 and TPR2 also interact with each other (Fig 9B). This suggests that the mechanism of TPR2 and TPR1 antagonism is based on titration of SNC1-TPR1 complexes by TPR2 or altered functions of a SNC1-TPR1-TPR2 complex.

## Discussion

To determine whether members of the *TPL* transcriptional repressor gene family functionally interact with *SRFR1* we chose a genetic approach. By creating double and higher order mutants between *srfr1-4*, members of the *TOPLESS* family, and other genes relevant to the *srfr1-4* autoimmune phenotype, we were able to assess the impact these genes had on constitutive immunity. Our results indicate a genetic interaction between *SRFR1* and *TPR2* and its close homolog *TPR3*. Further data show a novel genetic interaction between *SNC1* and *TPR2*. We found that stunting in *srfr1-4* was affected by mutations in *TPL* and *TPR2*, but in opposite ways; *srfr1-4 tpl-8* was less stunted, and *srfr1-4 tpr2-2* was more stunted. To verify that these phenotypes were a consequence of altered immune system regulation, and not a developmental phenotype unrelated to defense, we measured the expression of *PR2* as a marker of the defense response [31,32]. Previous research has shown that *PR1* and *PR2* mRNA levels are elevated in *srfr1-4* relative to wild type plants [12]. Here, we found that *PR2* levels in *srfr1-4 tpl-8* and *srfr1-4 tpr2-2* are indeed consistent with differentially regulated immune system outputs in these double mutants.

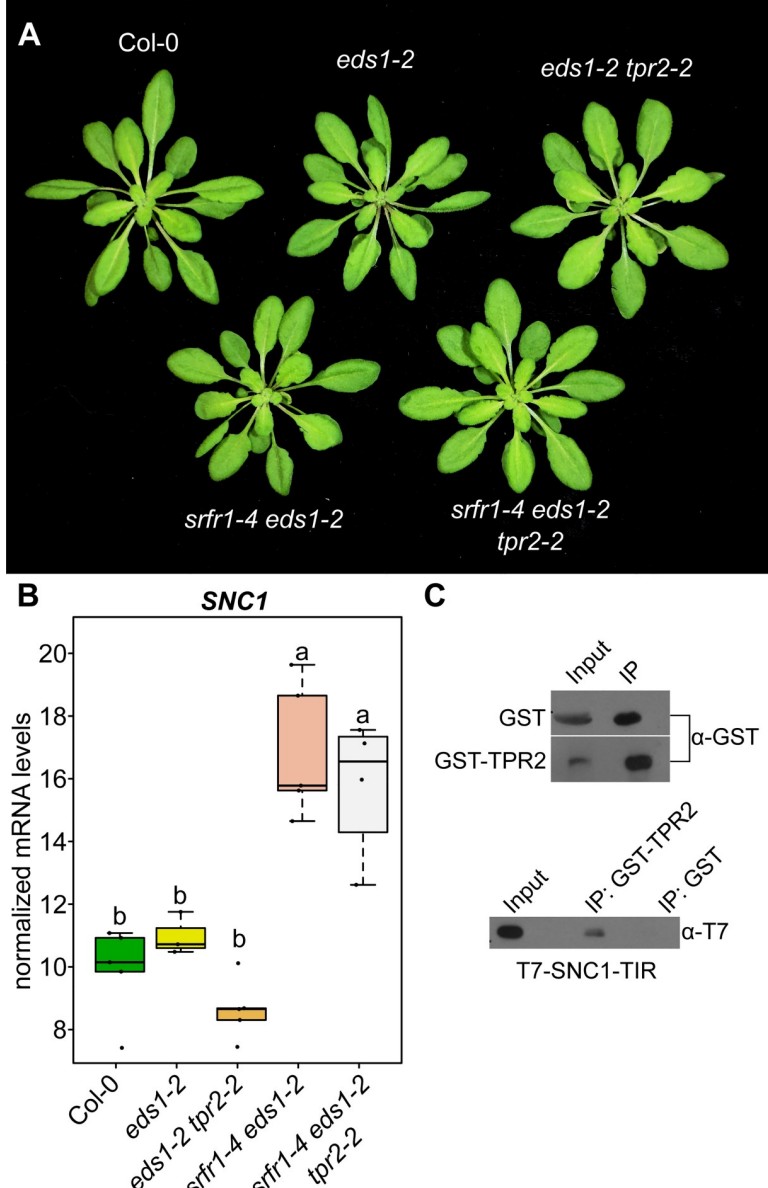

**Fig 8. SRFR1 acts upstream of SNC1 transcription.** (A) Morphological phenotypes of single, double, and triple mutants of *eds1-2*, *srfr1-4*, and *tpr2-2*. Plants were grown under short day conditions at 21˚C for four weeks. (B) Expression as measured by quantitative RT-PCR of *SNC1*. Dots represent individual data points taken over two separate experiments. Genes of interest were normalized against *SAND* (At2g28390). Whiskers on boxplots are drawn to the farthest data point within 1.5 * IQR of first and third quartiles. Letters denote significant differences as determined by Student's t-test (*P*<0.01) using the Bonferroni-Holm method to correct for multiple comparisons. (C) *In vitro* interaction of TPR2 and the TIR domain of SNC1 in *E. coli*. Proteins were pulled down and subjected to immunoblot analysis with either GST or T7 antibodies. This experiment was repeated once with similar results.

## Contrasting roles of *TPR1/TPL* and *TPR2/TPR3*

Stunting, but not all aspects of heightened basal resistance in *srfr1-4* has been previously shown to be dependent upon the TNL gene *SNC1* [12]. One mechanism by which SNC1 activates the immune system was demonstrated to be through a protein interaction with TPR1, the end result of this interaction being the repression of negative regulators of defense such as

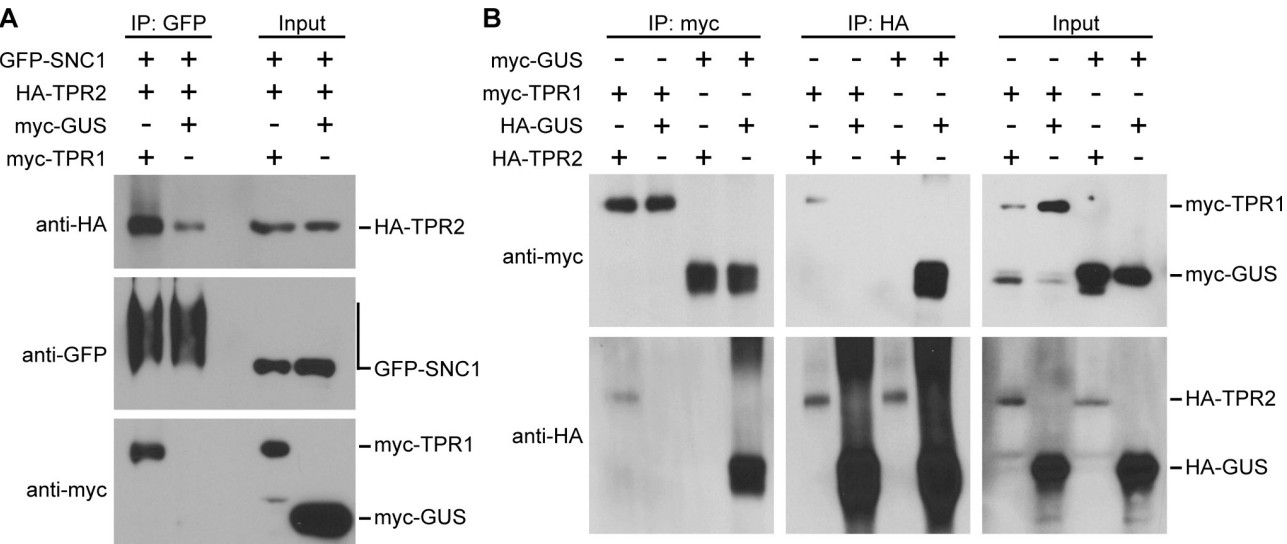

**Fig 9. SNC1, TPR1 and TPR2 interact with each other *in vivo*.** Tagged proteins were expressed in *N. benthamiana eds1-1* plants. (A) Immunoprecipitation of GFP-SNC1 and detection of co-immunoprecipitated myc-TPR1, and of HA-TPR2 in the presence or absence of myc-TPR1, with myc-GUS as a control. (B) Reciprocal co-immunoprecipitation of myc-TPR1 and HA-TPR2, with correspondingly tagged GUS as negative (interaction with TPR) and positive (self-interaction) control. The expected protein band positions based on molecular size markers are indicated to the right of blots.

*DND1* and *DND2*. *SNC1* was also shown to interact genetically with *TPL*, which shares 92% identity with TPR1 at the amino acid level [19]. The attenuated autoimmunity we observed in *srfr1-4 tpl-8* is in agreement with this model. We did not see a similar phenotype in *srfr1-4 tpr1-2*, most likely because the *tpr1-2* allele is not a true knockout. We verified by sequencing out from the T-DNA that the location of the *tpr1-2* insertion is within the first intron of *TPR1*, which is located in the 5' untranslated region. This insertion may not be sufficient to knock out transcription of functional *TPR1* mRNA.

In contrast to *srfr1-4 tpl-8*, the *srfr1-4 tpr2-2* phenotype is a novel case wherein a member of the *TOPLESS* family is implicated in repressing an immune response. Based on the strikingly different phenotypes of the double mutants we propose that TPR2 is repressing a set of genes disparate from that of TPR1 or is activating genes in the *srfr1-4* background. We verified that the exacerbated autoimmune phenotype in *srfr1-4 tpr2-2* was linked to *TPR2* by demonstrating that another allele of *TPR2*, *tpr2-1*, could produce the same phenotype in *srfr1-4*.

Previous research has shown varying degrees of redundancy amongst the different members of the *TOPLESS* family depending on the process under study. In embryogenesis and circadian clock regulation, knocking out all *TPL/TPR* genes is required to see a phenotype [21,33], whereas, for partial alleviation of repression of brassinosteroid-sensitive genes via BZR1, the *tpl tpr1 tpr4* triple mutant was sufficient [23]. Mutants of *TPR2* and *TPR3* were not included in this analysis and the partial nature of the *tpl tpr1 tpr4* mutant phenotype was assumed to be based on the presence of functional *TPR2* and *TPR3* [23]. A certain degree of redundancy may also explain the relatively weak effects of the majority of *tpr* single mutants on *srfr1-4*-mediated phenotypes. In addition, *TPR3*, the closest homolog of *TPR2*, has some functional redundancy with *TPR2* in repressing autoimmunity in *srfr1-4* in that the *srfr1-4 tpr2-2 tpr3-1* triple mutant is significantly more stunted than *srfr1-4 tpr2-2* and shows increased *PR2* levels relative to *srfr1-4 tpr2-2* and *srfr1-4*. It is a common observation that gene family members of transcriptional regulators display redundancy and that subsets of members effect opposite regulation, as in the cases of WRKY and TCP transcription factors in plant

immunity [34]. However, the context-specific function of TPR2/TPR3 as either redundant with or opposite to TPL/TPR1/TPR4 depending on the regulatory pathway is surprising.

## Contributions to *SNC1* regulation by *SRFR1*

Although stunting in *srfr1-4* is fully dependent upon *SNC1*, *SRFR1* has a broader effect on immune function independent of *SNC1*. The TNL resistance genes *RPS4*, *RPP4*, and At4g16950 are all upregulated in *srfr1* mutants independent of *SNC1*, as well as several other genes related to immune function such as *EDS1*, *PAD4*, *SID2*, *PR1*, and *PR2* [11,12]. *SNC1* is located within the *RPP5* disease resistance locus, a complex locus containing several paralogous resistance genes [35]. It has been previously shown that activation of SNC1 leads to increased transcription of other resistance genes at this locus, such as *RPP4* and *At4g16950* [12,24,36]. The mechanism by which *RPP4* and *At4g16950* are upregulated by activated SNC1 is unknown, although two possibilities were proposed in Yi and Richards. The first involves upregulation as a result of increased SA caused by SNC1 activation, citing previous work showing that application of SA is sufficient to cause a large increase in *SNC1* transcript [26]. However, they also do not rule out the possibility that chromatin structure at the locus might be altered due to increased transcription of *SNC1*, creating a permissive environment for transcription of neighboring paralogs [24].

Interestingly, *RPP4* and *At4g16950* are both upregulated in *srfr1-4 snc1-11* [12], a genetic background without a functional copy of *SNC1*, and as a consequence of this observation we hypothesized that the *PR2* increase we observed in *srfr1-4 snc1-11 tpr2-2 tpr3-1* could be due to a further increase in transcript of these other *RPP5* locus resistance genes. Surprisingly, *RPP4* levels were significantly decreased by adding the *tpr2* and *tpr3* mutations to *srfr1-4 snc1-11*, implying that the increased *RPP4* in *srfr1-4 tpr2-2* relative to *srfr1-4* is fully dependent upon increased *SNC1*. We therefore asked if *TPR2* had a genetic interaction with *SNC1* by crossing *tpr2-2* with *snc1-1*. The *snc1-1* allele contains a point mutation in the linker region between the NBS and LRR domains that causes constitutive activation of the SNC1 protein and associated stunting caused by induction of the defense response without increasing the levels of *snc1-1* mRNA [25,37]. In the *snc1-1 tpr2-2* double mutant we saw significantly increased stunting, and *snc1-1* and *PR2* mRNA levels, suggesting a role for *TPR2* in the downregulation of the SNC1-mediated constitutive defense response.

In order for resistance genes of the TNL class to function, the lipase like protein EDS1 must be present [38–40]. To elucidate the position of *TPR2* in the *SNC1*-mediated constitutive defense response we took advantage of the *srfr1-4 eds1-2* double mutant which blocks increased basal resistance in *srfr1-4* [14] and consequently feedback upregulation of *SNC1*. Other studies have used mutations in *EDS1*, and closely related protein interactor *PAD4* which is also required for *SNC1* signaling, to block feedback upregulation of *SNC1* to determine if genes are acting upstream or downstream of SNC1 activation [25,26,28,41]. In the *srfr1-4 eds1-2 tpr2-2* triple mutant we did not see a significant increase in *SNC1* mRNA absent of SNC1 protein activation compared to *srfr1-4 eds1-2*. This result implies that *TPR2* is acting downstream of SNC1 activation, whereas *SRFR1* also impacts the level of *SNC1* mRNA. This difference may be one component for the additive effect of mutations in *SRFR1* and *TPR2* on the level of constitutively activated defenses.

## Model for TPR2/TPR3 and SRFR1 functions in SNC1-mediated autoimmunity

Based on these data we present the following model for TPR2 and SRFR1 function in autoimmunity caused by SNC1 activation (Fig 10). In the *srfr1-4* background *SNC1* mRNA is

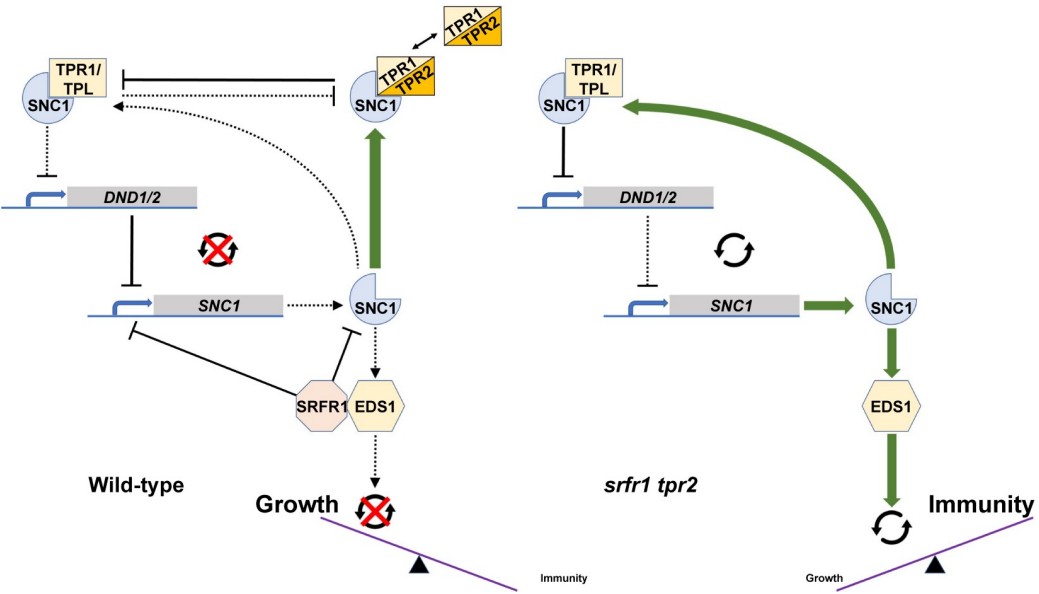

**Fig 10. Model for TPR2 and SRFR1 functions in SNC1-mediated autoimmunity.** (Left) In Col-0, low levels of *SNC1* help to avoid fitness penalties. This may be accomplished both through direct inhibition by SRFR1 and through sequestration by TPR2 of TPR1 to reduce TPR1-SNC1 interactions that affect negative regulators of immunity such as *DND1/DND2* and indirectly subsequent *SNC1* expression, or through formation of a SNC1-TPR1-TPR2 complex with altered functions compared to an inhibitory SNC1-TPR1 complex. Here, the combined effects of SRFR1 and TPR2 hold SNC1 expression in check. (Right) In the *srfr1-4 tpr2-2* double mutant, these molecular check points are released, allowing SNC1 expression to trigger an autoimmune response that results in excessive stunting.

expressed at a high level and SNC1 is constitutively activated [12]. Disruption of protein-protein interactions between SRFR1 and SNC1 [12] could lead to SNC1 activation; however, increased mRNA levels can also lead to SNC1 auto-activation [24,42] and based on SRFR1's interaction with TCP transcription factors a direct regulation of *SNC1* transcript levels [15,43] is consistent with the data obtained in the *eds1-2* background. Because in wild type plants levels of *SNC1* are kept low to avoid fitness penalties, the effects of *TPR2* mutations are only apparent when *SNC1* transcription is induced, such as in the autoimmune mutants *srfr1-4* and *snc1-1*. We hypothesize that TPR2, and to some degree TPR3, acts downstream of *SNC1* transcription by repressing expression of a positive regulator of *SNC1* or activating a negative regulator. Parsimoniously, the physical interaction of TPR2 with the TIR domain of SNC1 shown here raises the possibility that TPR2 competes with TPR1 for binding of SNC1, and that TPR1-SNC1 and TPR2-SNC1 complexes regulate target genes such as *DND1* and *DND2* in opposite ways.

Interestingly, *in vivo* data provide evidence for an alternative model in which the interaction between TPR1 and TPR2 leads to either sequestration of TPR1 to reduce formation of SNC1-TPR1 complexes or to sequestration of heteromeric SNC1-TPR1-TPR2 complexes with altered inhibitory functions. Given the context-specific degree of redundancy in the *TPL* family, it is worth emphasizing that the model developed here is perhaps limited to immune phenotypes or more narrowly to consequences of interactions with SNC1. Interestingly, members of the TPL-related metazoan Groucho/Transducin-Like Enhancer of split (Gro/TLE) transcriptional co-repressor family are known to be regulated by context-specific parameters, such as the levels of available partner repressors, post-translational protein modifications, or competition with activators [44,45].

In this regard, the recent demonstration of TPR1 regulation by SUMOylation is noteworthy [20]. SUMOylation reduces TPR1 repressor activity, resulting in increased expression of genes such as *DND1* and *DND2* and a dampened immune activation. In contrast to expectations, a version of TPR1 that cannot be SUMOylated interacts less with SNC1, suggesting that strong association of SUMOylated TPR1 with SNC1 leads to a sequestering of the co-repressor complex. A SNC1-TPR1-TPR2 heteromer may therefore either precipitate SUMOylation of TPR1 or mimic the inhibited SNC1-TPR1 co-repressor complex through structural differences between TPR1 and TPR2. It is perhaps relevant that we observed stronger *in vivo* interaction of TPR2 with SNC1 when TPR1 was present, although this requires more careful quantification.

Finally, enhancement of the *snc1-1* phenotype by *tpr2-2* illustrates that the enhanced resistance phenotype is not dependent upon mutations in *SRFR1*. Together, this suggests that TPR2 and SRFR1 are involved in separate pathways converging on regulation of SNC1. The mechanism of TPR2's regulation of *SNC1*-mediated autoimmunity merits further study.

## Materials and methods

### Plant lines

Plant lines used for genetic analysis were *tpl-8* (SALK_036566), *tpr1-2* (SALK_065650C), *tpr2-1* (SALK_112730), *tpr2-2* (SALK_079848C), *tpr3-1* (SALK_029936), *tpr4-1* (SALK_150008), *snc1-11* (SALK_047058) from the Salk T-DNA knockout collection [46]. The *srfr1-4* line (SAIL_412-E08) was from the Syngenta Arabidopsis Insertion Library [47]. Salk and SAIL lines were acquired from the Arabidopsis Biological Resource Center. The *eds1-2* line was a gift from Jane Parker, and the *snc1-1* line was a gift from Harrold van den Burg. All mutants are in the Col-0 background, and genotyping primers used for these lines are detailed in S1 Table. After parental lines were crossed, plants were genotyped in the F1 generation to verify the success of the cross, and then in the F2 generation to identify plants homozygous for the desired mutations. For plant growth and *in planta* bacterial growth assays, plants were grown in environmentally controlled conditions (Controlled Environments Ltd., Winnipeg, Manitoba, Canada; 8 h light / 16 h dark, 90–140 µmol photons m$^{-2}$ s$^{-1}$; 21°C, 70% humidity). Strain DC3000 of *P. syringae* was infiltrated into leaves of 4 week-old plants with a needle-less syringe at a bacterial density of $5\times10^4$ colony-forming units per ml resuspended in 10 mM MgCl$_2$. Tissue samples were collected at day 0 and day 3 post inoculation and analyzed as described previously [12].

### Molecular cloning and generation of transgenic lines

The TPR2-myc construct was created by amplifying the *TPR2* CDS with flanking SpeI and PacI sites at the 5' and 3' ends, respectively. The binary vector pGWB20 [48] was cut with XbaI and PacI to excise the Gateway cassette, and the SpeI-*TPR2*-PacI fragment was ligated into the XbaI and PacI sites in frame with the C-terminal myc tags in pGWB20. Sequencing was used to verify the clone. *Agrobacterium tumefaciens* strain C58-C1 was transformed with the TPR2-myc construct by electroporation. The *srfr1-4 tpr2-2* double mutant was grown at high temperatures to relieve stunting, and these plants were transformed by floral dip. Transgenic seed was selected on hygromycin B, and T3 homozygotes were selected by true breeding on selection plates. TPR2-myc protein expression was verified by western blot using c-Myc antibody sc-789 (Santa Cruz Biotechnology, Dallas, TX, USA).

The GST-TPR2 construct was created by amplifying the *TPR2* CDS with flanking EcoRI and NotI sites with an additional base between the EcoRI site and the start codon. The EcoRI-*TPR2*-NotI fragment was cloned into pGEX-4T-3 (SigmaAldrich, St. Louis, MO, USA)

digested with EcoRI and NotI. Similarly, a cDNA encoding the SNC1 TIR domain (amino acids 1–182) was amplified with flanking EcoRI and XhoI sites. The EcoRI-*TIR*-XhoI fragment was cloned into pET28a (EMD Millipore, Billerica, MA USA) digested with EcoRI and XhoI to create *His-T7-SNC1 TIR*.

## RNA extraction, cDNA preparation and qPCR

For qPCR experiments multiple plants from each genotype were ground together in liquid nitrogen to form one replicate. For each experiment two or three replicates were used per genotype. After grinding plant tissue in liquid nitrogen, total RNA was extracted using TRI-ZOL reagent (Thermo Fisher Scientific, Carlsbad, CA, USA). First strand cDNA synthesis was carried out using 2 μg of total RNA and reverse transcription was performed using an oligo (dT) 15 primer and Moloney murine leukemia virus (MMLV) reverse transcriptase (Promega, Madison, WI, USA). qPCR was carried out using SYBR GREEN PCR Master Mix (Thermo Fisher Scientific) or Brilliant III Ultra-Fast SYBR Green qPCR Master Mix (Agilent, Santa Clara, CA, USA) on either an ABI 7500 or Agilent AriaMX qPCR system. Transcript levels were normalized using *SAND* gene (At2g28390) for qPCR experiments. LinRegPCR was used to determine PCR efficiency and cycle thresholds for each sample [49], and the $2^{-\Delta\Delta C_T}$ method was used to determine expression levels [50]. Primers used for qPCR are detailed in S2 Table.

## Protein pull-down and co-immunoprecipitation assays

GST-TPR2, empty pGEX-4T-3, and T7-SNC1-TIR in *E. coli* strain BL21(DE3) were streaked to single colonies and then incubated overnight at 37˚C in LB broth. 200 ml of LB was inoculated with 2 ml of overnight culture and incubated for approximately 3 hours to an optical density of 0.6–0.8. IPTG at 500 μM was added to each culture and flasks were grown overnight at 22˚C. Each culture was passed through a French press to lyse the cells. Extracts were centrifuged and 25 μl of GST beads (G-Biosciences, St. Louis, MO USA) were added to 6 μl supernatant of GST-TPR2 and empty pGEX-4T-3. Samples were incubated at 4˚C for 1.5 hours with rotation. After washing 3 times with PBS, 6 μl soluble protein T7-SNC1-TIR was added, and samples were incubated at 4˚C for 1 hour. After washing 3 times with PBS protein was eluted from beads in Laemmli buffer and then used for protein blot with anti-GST and anti-T7 (EMD Millipore). For PR2 detection in S2 Fig, PR2 antibody AS207 208 (Agrisera, Vannas, Sweden) was used.

For co-immunoprecipitation (co-IP) assays, agrobacterium strain C58C1 containing the corresponding constructs was infiltrated into *N. benthamiana eds1-1* plants [30]. Infiltrated leaf areas were harvested two days post infiltration. Co-IP was performed as described by [51]. In brief, 3 g fresh tissue were ground into fine powder in liquid nitrogen and solubilized with 9 ml protein extraction buffer (100 mM Tris-HCl pH 7.5, 300 mM NaCl, 2 mM EDTA pH 8.0, 1% Triton X-100, 10% glycerol, and protease inhibitor). Protein extracts were centrifuged twice at 14,800 rpm for 15 min. Supernatants were incubated with anti-HA (Sigma, E6779), anti-myc (Sigma, E6654) or anti-GFP (MBL, D153-8) beads at 4˚C for 90 min. After precipitation agarose beads were washed three times with a washing buffer (50 mM Tris-HCl pH 7.5, 150 mM NaCl, 1 mM EDTA pH 8.0, 0.5% Triton X-100, 5% glycerol, and protease inhibitor). Beads were then boiled with 1×SDS loading buffer. Tagged proteins were detected with anti-HA (Roche, 12013819001), anti-myc (Roche, 11814150001) or anti-GFP (Invitrogen, A11122) antibodies.

## Statistical analyses

Student's t-test with the Bonferroni-Holm method to correct for multiple comparisons was used for all statistical analyses. Raw data underlying the analyses can be found in S1 Data for

data represented in main figures, in S2 Data for shoot fresh weights in S1 Fig, and in S3 Data for *in planta* bacterial growth data in S6 and S7 Figs.

## Supporting information

**S1 Fig. Shoot weights of *tpl/tpr* single mutants do not differ significantly from Col-0.** Shoot weight from plants grown under short day conditions at 21˚C for four weeks. Dots represent individual data points taken over two separate experiments. Whiskers on boxplots are drawn to the farthest data point within 1.5 * IQR of first and third quartiles. Letters denote significant differences as determined by Student's t-test (P<0.01) using the Bonferroni-Holm method to correct for multiple comparisons.
(PDF)

**S2 Fig. PR2 expression in *srfr1-4* is affected by *tpl* and *tpr2*.** Western blot of total protein extracted from *srfr1-4*, *srfr1-4 tpl-8*, *srfr1-4 tpr1-2*, *srfr1-4 tpr2-2*, *srfr1-4 tpr3-1*, and *srfr1-4 tpr4-1*. The large subunit of rubisco is shown as a loading control.
(PDF)

**S3 Fig. Comparison of stunting in *srfr1-4 tpl/tpr* double mutants at 28$^0$ and 21˚C.** Growth phenotype of *srfr1-4* and *srfr1-4 tpl/tpr* double mutants at 28$^0$ (top row) and 21˚C (bottom).
(PDF)

**S4 Fig. Molecular characterization of *tpr2* T-DNA insertion alleles.** (A) T-DNA insertion locations for *tpr2-1* (SALK_112730) in exon 13, and *tpr2-2* (SALK_079848) in exon 21. Scale bar is 200 bp. (B) RT-PCR using *TPR2* primers on the 3' side of the T-DNA insertions in *tpr2-1* and *tpr2-2* after 33 cycles. *Actin* was used as a control for quality of RNA and efficiency of reverse transcription.
(PDF)

**S5 Fig. Phylogenetic tree of the Arabidopsis TOPLESS family.** Phylogram showing evolutionary relationships amongst *TOPLESS* family members. The WD40 protein *LEUNIG* (*LUG*) is included as the outgroup. Tree was generated from full length cDNA sequences using www.phylogeny.fr.
(PDF)

**S6 Fig. Slightly elevated *SNC1* expression in *tpr2-2 tpr3-1* plants does not lead to increased bacterial resistance.** *In planta* bacterial growth assay with the indicated plant genotypes and DC3000 infiltrated at a bacterial density of $5\times10^4$ colony-forming units (cfu) per ml. Values represent averages from two independent experiments with triplicate samples, and error bars denote standard deviation. As determined by Student's t-test with the Bonferroni-Holm method to correct for multiple comparisons, none of the values were significantly different with $P\geq0.2$.
(PDF)

**S7 Fig. *PR2* expression levels correlate with degree of bacterial resistance.** *In planta* bacterial growth assay with the indicated plant genotypes and DC3000 infiltrated at a bacterial density of $5\times10^4$ cfu/ml. Values represent averages from two independent experiments with triplicate samples, and error bars denote standard deviation. Letters denote statistically significant differences as determined by Student's t-test with the Bonferroni-Holm method to correct for multiple comparisons (*P*<0.05).
(PDF)

**S1 Table. PCR primers used for genotyping mutant lines.**
(PDF)

**S2 Table. Primers used for qPCR.**
(PDF)

**S1 Data. Raw data file for main figures.**
(XLSX)

**S2 Data. Raw data file for shoot weights shown in S1 Fig.**
(XLSX)

**S3 Data. Raw data file for *in planta* bacterial growth assays in S6 and S7 Figs.**
(XLSX)

## Acknowledgments

We thank Harrold van den Burg for the *snc1-1* plant line, Brian Staskawicz for *N. benthamiana eds1-1* seed, Gary Stacey for the pGWB20 vector, and Daniel Leuchtman and Sanzida Rahman for help with statistical analyses.

## Author Contributions

**Conceptualization:** Christopher M. Garner, Walter Gassmann.

**Formal analysis:** Christopher M. Garner, Benjamin J. Spears.

**Funding acquisition:** Christopher M. Garner, Walter Gassmann.

**Investigation:** Christopher M. Garner, Benjamin J. Spears, Jianbin Su, Leland J. Cseke, Samantha N. Smith, Conner J. Rogan.

**Methodology:** Christopher M. Garner, Benjamin J. Spears, Jianbin Su.

**Project administration:** Christopher M. Garner, Walter Gassmann.

**Visualization:** Christopher M. Garner, Leland J. Cseke.

**Writing – original draft:** Christopher M. Garner.

**Writing – review & editing:** Christopher M. Garner, Leland J. Cseke, Walter Gassmann.

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
