## [Decision Letter · Decision Letter 0]

26 Aug 2020

Dear Dr Gassmann,

Thank you very much for submitting your Research Article entitled 'Opposing functions of the plant TOPLESS gene family during SNC1-mediated autoimmunity' to PLOS Genetics. Your manuscript was fully evaluated at the editorial level and by independent peer reviewers. The reviewers appreciated the attention to an important problem, but raised some substantial concerns about the current manuscript. Based on the reviews, we will not be able to accept this version of the manuscript, but we would be willing to review again a revised version. We cannot, of course, promise publication at that time.

Should you decide to revise the manuscript for further consideration here, your revisions should address the specific points made by each reviewer. We will also require a detailed list of your responses to the review comments and a description of the changes you have made in the manuscript. In particular, please focus your efforts on infection assays and the role of TPR2 and TPR3 with respect to SNC1. These points were made by all reviewers, and I agree that they should be addressed. 

If you decide to revise the manuscript for further consideration at PLOS Genetics, please aim to resubmit within the next 60 days, unless it will take extra time to address the concerns of the reviewers, in which case we would appreciate an expected resubmission date by email to plosgenetics@plos.org.

[LINK]

We are sorry that we cannot be more positive about your manuscript at this stage. Please do not hesitate to contact us if you have any concerns or questions.

Yours sincerely,

Gitta Coaker, PhD

Associate Editor

PLOS Genetics

Gregory P. Copenhaver

Editor-in-Chief

PLOS Genetics

Reviewer's Responses to Questions

**Comments to the Authors:**

Reviewer #1: Garner et al. describe the negative role of TOPLESS family members in SRFR1/SNC1-dependent immunity. Using plant dwarf morphology as a proxy to defense autoactivation, they performed epistasis analysis between srfr1 and tpr mutants, looking for enhancement or reversion of srfr1 phenotype. They demonstrated that trp2 and tpr3 loss-of-function mutant both enhance srfr1/snc1 stunting growth and defense gene expression. Whereas, TPR2 overexpression suppressed srfr1 and srfr1 tpr2 growth. They further showed that srfr1 tpr2 tpr3 phenotypes depend on SNC1. Coupled with gene expression analysis, the authors claimed that SRFR1 functions upstream of SNC1 transcription, while TPR2, on the other hand, may act downstream.

The manuscript is overall convincing and easy to follow. The genetic data is solid and clear; however, infection assays and mechanisms are lacking. To justify publication in PLOS Genetics, more characterization of the mutants and some mechanism are needed:

Major comments:

- Infection assays: why the authors did not perform a single infection assay for the mutants? Plant size and marker gene expression are helpful. But without bona fide pathogen challenges, it is insufficient to draw immune-related conclusions. It is reasonable bacterial infections are difficult to be done on dwarf mutants, but not wild-type like plants.

- The role of TPR2/3 on SNC1 is unclear. At least some molecular mechanism should be provided.

Minor comments:

- It would be helpful to place parental lines in the same images:

Fig3A, srfr1 tpr3 double mutant

Fig4A, tpr2, tpr3 single mutant

- L140: to claim tpr1-2 is not a true knockout need to show T-DNA genotyping and RNA level.

- L166, it is insufficient to claim tpr2 tpr3 act redundantly to enhance srfr1. Under autoimmune mutant background, this could also due to additive effects of independent pathways.

- L168, may worthwhile performing infection assays on tpr2 tpr3 double mutants. Certain mutant with mild EDR phenotype may not exhibit obvious growth defects.

- L232: should it be “decrease” in SNC1 expression?

- L240: no evidence supports TPR2 compete with TPR1

- Fig8C, in vitro assays are fine for protein interactions. I would like to see at least an additional in vivo IP to support the claim. Transient expression in tobacco would be sufficient.

- Fig9: Without additional data the model depicting competition between TPR2/SNC1 and TPR1/SNC1 cannot be made. These can also be tested via transient expression in tobacco.

Reviewer #2: The work is beautiful and the genetic data to reveal the interaction between SRFR1 and TPR2 and TPR3 is clearly present. However, about the function of TPR2 and TPR3 in negatively regulating plant autoimmunity, the conclusions are not that convincing, and the proposed mechanisms are a little speculative. Additive experiments are recommended to confirm the conclusions and strengthen the hypothesis.

Major

1.The authors claim a novel role of TPR2 and TPR3 in suppressing plant autoimmunity, pathogen assays are recommended to proof such a statement. PstDC3000 (virulent) or Pst DC3000 AvrRps4 (avirulent) strains could be used to test the srfr1-4 tpr2-2 and srfr1-4 tpr2-2 tpr3-1 mutants to provide further information on the effect of TPR2 and TPR3 on pathogen resistance.

2. The authors propose that TPR2 and to some degree TPR3 act downstream of SNC1 transcription by repressing expression of a positive regulator of SNC1 or activating a negative regulator. RNA-seq of srfr1-4, srfr1-4 tpr2-2, and srfr1-4 tpr2-2 tpr3-1 mutants are recommended, which could better explain the diverse function of these two TOPLESS members.

Minor

1. In figure 1, the letters that denote the significant differences are placed on top of the figure, which is difference from the letter placements in other figures. It is better to be consistent.

2. In page 5, line 85 “PR genes” should be “PR genes”, “PR” should be in italics.

3. In page 9, line 171 “Fig 4C and 4D” should be “Fig 4B and 4C”.

Reviewer #3: please see attachment

**Have all data underlying the figures and results presented in the manuscript been provided?**

Reviewer #1: Yes

Reviewer #2: Yes

Reviewer #3: Yes

PLOS authors have the option to publish the peer review history of their article (what does this mean?). If published, this will include your full peer review and any attached files.

Reviewer #1: No

Reviewer #2: No

Reviewer #3: No

---

## [Decision Letter · Decision Letter 1]

21 Jan 2021

Dear Dr Gassmann,

Thank you very much for submitting your Research Article entitled 'Opposing functions of the plant TOPLESS gene family during SNC1-mediated autoimmunity' to PLOS Genetics.

The manuscript was fully evaluated at the editorial level and by independent peer reviewers. The reviewers appreciated the attention to an important topic but identified some concerns that we ask you address in a revised manuscript

We therefore ask you to modify the manuscript according to the review recommendations. Your revisions should address the specific points made by each reviewer. All reviewer comments can be textually addressed and do not require new experimentation. In particular, reviewer 1 suggests addressing redundancy in TPL/TPR loci as well as consolidating some of the main and supplemental figures. I agree with the comment about TPL/TPR loci and leave the decision of figure consolidation to your discretion. 

[LINK]

Yours sincerely,

Gitta Coaker, PhD

Associate Editor

PLOS Genetics

Gregory P. Copenhaver

Editor-in-Chief

PLOS Genetics

Reviewer's Responses to Questions

**Comments to the Authors:**

Reviewer #1: The authors have addressed most of my concerns. This reviewer has a couple of final suggestions:

1. As a dominant-negative mutant of topless tpl-1 is lethal, it is predicted that knocking out all five TPL/TPR related genes would yield a lethal phenotype. Therefor the whole TPL family must be redundant in many aspects. This may explain the weak phenotypes of the mutants the authors describe. The new interaction data between TPR1 and TPR3 can also support such redundancy since they are exchangeable in the complex. Some discussion should be added regarding this, so the readers would bear in mind the diverse, and sometimes even opposite roles of these repressor proteins in regulating plant biology. An alternative explanation for the story can be that the TPL family proteins all serve both positive and negative roles, the previous 3 more positive, while 2/3 more negative. I would also suggest the authors to include such alternative model in their model figure.

2. Some main figures are quite thin, and there are too many figures/sup figures. I would suggest the authors to reduce the number of figures below 7, combining some simples ones but include some of the important supplementary figures in main.

Reviewer #2: In the revised manuscript, the authors added data on planta bacterial growth assays. Their conclusions about the role of TPR2 and TPR3 in suppressing plant autoimmunity are more solid and convincing now with the addition of the new generated data. Most of my previous concerns have been well addressed.

One of my previous concerns was not entirely addressed: although the authors showed that TPR2 directly interacts with SNC1, the mechanism on how this interaction represses negative regulators of immunity is still not quite clear.

Since the main focus of this manuscript is to establish the functional relationship between SRFR1 and TPR, it is acceptable that the authors did not look into the detailed differences in gene regulation by RNA-Seq in this manuscript.

Reviewer #3: My concerns have been sufficiently addressed.

**Have all data underlying the figures and results presented in the manuscript been provided?**

Reviewer #1: Yes

Reviewer #2: Yes

Reviewer #3: Yes

PLOS authors have the option to publish the peer review history of their article (what does this mean?). If published, this will include your full peer review and any attached files.

Reviewer #1: No

Reviewer #2: No

Reviewer #3: No

---

## [Editor Report · Decision Letter 2]

5 Feb 2021

Dear Dr Gassmann,

We are pleased to inform you that your manuscript entitled "Opposing functions of the plant TOPLESS gene family during SNC1-mediated autoimmunity" has been editorially accepted for publication in PLOS Genetics. Congratulations!

Yours sincerely,

Gitta Coaker, PhD

Associate Editor

PLOS Genetics

Gregory P. Copenhaver

Editor-in-Chief

PLOS Genetics

Comments from the reviewers (if applicable):

**Data Deposition**

http://datadryad.org/submit?journalID=pgenetics&manu=PGENETICS-D-20-01202R2

**Press Queries**

---

## [Editor Report · Acceptance letter]

18 Feb 2021

PGENETICS-D-20-01202R2 

Opposing functions of the plant *TOPLESS* gene family during SNC1-mediated autoimmunity 

Dear Dr Gassmann, 

We are pleased to inform you that your manuscript entitled "Opposing functions of the plant *TOPLESS* gene family during SNC1-mediated autoimmunity" has been formally accepted for publication in PLOS Genetics! Your manuscript is now with our production department and you will be notified of the publication date in due course.

With kind regards,

Alice Ellingham

PLOS Genetics

On behalf of:
